# Defect of IL17 Signaling, but Not Centrinone, Inhibits the Development of Psoriasis and Skin Papilloma in Mouse Models

**DOI:** 10.3390/biomedicines10081976

**Published:** 2022-08-15

**Authors:** Ben Jin, Yongfeng Zhang, Haiyan D. Miller, Ling He, Dongxia Ge, Alun R. Wang, Zongbing You

**Affiliations:** 1Southeast Louisiana Veterans Health Care System, New Orleans, LA 70119, USA; 2Department of Structural & Cellular Biology, Tulane University, New Orleans, LA 70112, USA; 3Department of Orthopaedic Surgery, Tulane University, New Orleans, LA 70112, USA; 4Department of Pathology and Laboratory Medicine, Tulane University, New Orleans, LA 70112, USA; 5Tulane Cancer Center and Louisiana Cancer Research Consortium, Tulane University, New Orleans, LA 70112, USA; 6Tulane Center for Stem Cell Research and Regenerative Medicine, Tulane University, New Orleans, LA 70112, USA; 7Tulane Center for Aging, Tulane University, New Orleans, LA 70112, USA

**Keywords:** psoriasis, skin papilloma, IL17RA, IL17RC, centrinone

## Abstract

Patients with psoriasis tend to develop skin cancer, and the hyperproliferation of the epidermis is a histopathological hallmark of both psoriasis and cutaneous squamous cell carcinoma (SCC), indicating that they may share pathogenic mechanisms. Interleukin-17 (IL17) stimulates the proliferation of the epidermis, leading to psoriasis. Overexpression of Polo-like kinase 4 (PLK4), which controls centriole duplication, has been identified in SCC, which also shows the hyperproliferation of keratinocytes. To investigate the cooperation between IL17 signaling and centriole duplication in epidermal proliferation, we established psoriasis and skin papilloma models in wild type (WT), IL17 receptor A (T779A) knockin (*Il17ra*(T779A)-KI), and IL17 receptor C knockout (*Il17rc*-KO) mouse strains. Bioinformatics, Western blot, immunohistochemical staining, colony formation, and real-time PCR were used to determine the effect of IL17 signaling and centrinone on epithelial proliferation. In the psoriasis model, compared to WT and *Il17ra*(T779A)-KI, *Il17rc*-KO dramatically suppressed epidermal thickening. The proliferation of keratinocytes significantly decreased in this order from WT to *Il17ra*(T779A)-KI and *Il17rc*-KO mice. In the skin papilloma model, *Il17ra*(T779A)-KI significantly decreased tumor burden compared to the WT, while Il17rc-KO abolished papilloma development. However, centrinone, a selective inhibitor of PLK4, did not affect skin lesion formation in either model. Our data demonstrated that *Il17ra*(T779A)-KI and *Il17rc*-KO prevent the development of psoriasis and tumorigenesis in the skin, while the topical administration of centrinone does not have any effect.

## 1. Introduction

Psoriasis is a complex, chronic autoimmune disease accompanied by systematic symptoms [1]. The GBD 2019 study reported that newly diagnosed cases of psoriasis were approximate 4,622,594 worldwide [2]. In the U.S., around 7.55 million adults aged 20 years or older were estimated to have psoriasis in 2020 [3]. Multiple comorbidities, including cardiometabolic diseases, specific-site cancer, and mental disorders, gradually appear along with psoriasis [3]. A recent systematic review and meta-analysis revealed that patients suffering severe psoriasis had a significantly elevated incident risk to develop squamous cell carcinoma (risk ratio (RR), 11.74 [95% confidence interval (CI), 1.52–90.66]). Overall, a high incident risk was also found in patients of all severities of psoriasis (RR, 2.15 (95% CI, 1.32–3.5)) [4]. Cutaneous squamous cell carcinoma (SCC), with around 1.8 million new cases in the US every year, is the second most common form of non-melanoma skin cancer [5]. The aberrant proliferation of keratinocytes leads to the thickened epidermis in psoriasis [6] and SCC [7]. Due to the similar histopathological alterations, psoriasis and SCC may share pathogenic mechanisms, to some extent. Even though phototherapy of psoriasis was verified to increase SCC and basal cell carcinoma (BCC) [8], the specific molecular mechanisms of the link between psoriasis and SCC are still elusive.

Hyperproliferation and abnormal differentiation of epidermal keratinocytes are provoked by interleukin-23 (IL23) and interleukin-17 (IL17) signaling [9,10]. In psoriasis, IL23 secreted by dendritic cells induces Th17 cells and γδ T cells to produce IL17 [11]. Epidermal keratinocytes are the main target cells of IL17 in the skin. With the stimulation of IL17, keratinocytes hyper-proliferate and synthesize chemokines, such as C-X-C motif chemokine ligand 1 (CXCL1), C-C motif chemokine ligand 20 (CCL20) [6], and cytokines, such as interleukin-20 (IL20), interleukin-36 (IL36), and tumor necrosis factor α (TNF α) [12,13] to recruit neutrophils, macrophages, and other immune cells, leading to continuous inflammatory responses and skin lesions, including erythematosus, acanthosis, and parakeratosis. Although monoclonal antibodies that specifically target IL23, IL17, and IL17 receptor A (IL17RA) significantly improve psoriasis symptoms [1], the therapy cannot eliminate pathogenic T cells, which subsequently contribute to a relapse of psoriasis through again provoking keratinocytes [14,15]. Therefore, besides blocking IL17 signaling, an alternative approach is needed to inhibit the uncontrolled proliferation of keratinocytes. In SCC, excessive exposure to ultraviolet (UV) radiation and subsequent DNA damage are considered the main risk factors to provoke the hyperproliferation of keratinocytes [16]. Mutations of tumor suppressor protein p53 (TP53), retinoblastoma transcriptional corepressor 1(RB1), cyclin-dependent kinase inhibitor 2A(CDKN2A), notch receptor 1(NOTCH), and RAS genes (HRAS, NRAS, and KRAS) lead to unrestrained cell cycling and the uncontrolled proliferation of keratinocytes [17,18]. Though stromal responses involving CD4^+^, T cell-mediated allergic inflammation demonstrates the importance of the microenvironment in SCC development [19], but little is known about the role of IL17 signaling-mediated inflammatory responses in SCC. Due to the hyperproliferation of epidermal keratinocytes in these common skin diseases and the fact that the SCC risk is elevated by psoriasis, it is pivotal to find a new strategy to target the hyperproliferation of epidermal keratinocytes. Polo-like kinase 4 (PLK4) is a master regulator of centriole replication [20], and its overexpression has been identified in in vitro and ex vivo studies of basal cell carcinoma (BCC) and SCC of the skin [21,22]. As a key regulator of a conserved pathway regulating centriole biogenesis, PLK4 promotes centriole duplication in the G1/S transition, while self-phosphorylation prevents centriole reduplication in the S phase [23]. Since aberrant activation of IL17 signaling and centrosome amplification can enhance cellular proliferation, they have been speculated as potential targets in the treatment of psoriasis and skin cancer. However, the effects of targeting both IL17 signaling and centrosome amplification in skin lesions have not been determined.

Our previous study demonstrated that Threonine 780 of human IL17RA or Threonine 779 of mouse IL17RA were potential phosphorylation sites that meditate the degradation of IL17RA [24]. Heterodimer of IL17RA and IL17 receptor C (IL17RC) is recognized as the main receptor complex for IL17A and IL17F [25]. Therefore, mutation or knockout of either IL17RA or IL17RC may lead to the disturbance of IL17 signaling. We hypothesized that the inhibition of centriole duplication enhances the blockade of epidermal proliferation through damaging the heterodimerization of IL17RA/IL17RC. In the present study, to investigate potential cooperation between IL17 signaling and centriole duplication in the proliferation of the epidermis, we used wild type (WT) mouse strain with C57BL/6J background to generate a mouse strain with T779A mutation, IL17 receptor A (T779A) knockin (*Il17ra*(T779A)-KI), and an IL17 receptor C knockout (*Il17rc*-KO). Since psoriasis significantly increases the risk of SCC, a potential association between them is worth exploring. We used 69 mice to establish an imiquimod-induced psoriasis model and 37 mice to set up a classical two-stage model of skin carcinogenesis induced by 7,12-dimethylbenz[a]anthracene (DMBA) and 12-O-tetradecanoylphorbol-13-acetate (TPA). We found that the proliferation of keratinocytes in both models was not stalled by the PLK4 inhibitor centrinone, but was inhibited by *Il17rc*-KO. *Il17ra*(T779A)-KI significantly inhibited skin papilloma formation, but only slightly decreased epidermal thickening in the psoriasis model.

## 2. Materials and Methods

### 2.1. Mice

All procedures involving mice were approved by the Institutional Animal Care and Use Committee (IACUC) of Tulane University (Protocol# 1063). Littermate controls including both male and female mice were utilized to establish the models in this project. We maintained mice in specific-pathogen-free facilities at Tulane University. C57BL/6J WT mice were directly purchased from the Jackson Laboratory (Bar Harbor, ME). IL17 receptor A (T779A) knockin (*Il17ra*(T779A)-KI) strain (C57BL/6J genetic background) was custom-made at the Jackson Laboratory through clustered regularly interspaced short palindromic repeats and Cas9 (CRISPR/Cas9)-mediated genomic manipulation [26]. IL17 receptor C knockout (*Il17rc*-KO) strain (C57BL/6J genetic background) was generated in our lab, as described previously [27]. Mouse-tail DNA was isolated to confirm knockin and knockout mutations by polymerase chain reaction (PCR) and DNA sequencing.

### 2.2. Establishment of Psoriasis Model

The psoriasis model was established using imiquimod cream (Perrigo) to induce the proliferation of the epidermis in 8-week-old mice [28,29]. In brief, 1.25 mg of 5% imiquimod cream was smeared on the shaved back skin of WT, *Il17ra*(T779A)-KI, and *Il17rc*-KO mice for 5 consecutive days. On the right side of the back skin, 8 µM of centrinone (cat# HY-18682, MedChemExpress, Monmouth Junction, NJ, USA) dissolved in 200 µL of acetone (VWR) was topically administrated on the skin after smearing with imiquimod, while on the left side of the back skin, 200 µL of acetone was applied as a solvent control after smearing with imiquimod. After 2 days of consecutive imiquimod use, the mice were euthanized, and skin tissues were harvested and fixed for subsequent histological examination.

### 2.3. Establishment of a Two-Stage Model of Skin Papilloma

The 8-week-old mice received topical administration to establish a two-stage model of skin papilloma, according to the classical approach [30]. On the left and right back skin, tumor initiation was induced by 100 µM 7,12-dimethylbenz[a]anthracene (DMBA, cat# D3254, Millipore Sigma, St. Louis, MO, USA) dissolved in 200 µL of acetone. One week later, on both sides of the back skin, 75 µg/mL 12-O-tetradecanoylphorbol-13-acetate (TPA, cat# P1585, Millipore Sigma, St. Louis, MO, USA) in 100 µL of acetone was topically applied twice per week. To determine the effect of centrinone on the development of the tumor, 8 µM of centrinone dissolved in 200 µL of acetone was topically applied to the right side of shaved back skin, with 200 µL of acetone applied as a control on the left side. The treatment with centrinone was also conducted twice per week. The application of TPA and centrinone was continued for 13 weeks, and the status of each mouse and the appearance of tumors on the back skin were monitored weekly. Fourteen weeks after the first topical treatment of DMBA, the mice were euthanized, and the skin tissue was collected. Palpable nodules with a diameter equal to or greater than 1 mm were considered tumors and they were recorded as such. Tumor incidence was calculated as a percentage of mice bearing tumors. Tumor multiplicity was defined as the average number of tumors per mouse. The length, width, and depth of each tumor were measured by a caliper. Tumor volume was calculated by the modified formulas [31]:(1)if the depth is less than 1 mm, use
$$V=(\frac{W}{2}\times L)/2$$
(2)if the depth is equal to or greater than 1 mm, use
$$V=\frac{4}{3}\times \pi \times \frac{L}{2}\times \frac{W}{2}\times \frac{D}{2}$$
where *V* is the tumor volume, *W* is the tumor width, *L* is the tumor length, and *D* is the tumor depth.

### 2.4. Hematoxylin and Eosin Staining

Skin tissues were fixed with 4% paraformaldehyde overnight at room temperature and embedded into paraffin. Each tissue was consecutively sectioned into samples of 4 µm in thickness and mounted on glass slides for histology. Sections were baked at 60 °C for 1 h, and a series of deparaffined and hydration processes were conducted according to routine methods. Sections were incubated in CAT hematoxylin (cat# CATHE-MM, BIOCARE MEDICAL, Pacheco, CA, USA) for 3 min, 1× Tacha’s Blue solution (cat# HTBLU-MX, BIOCARE MEDICAL, Pacheco, CA, USA) for 3 min, and Edgar Degas Eosin solution (cat# HTE-GL, BIOCARE MEDICAL, Pacheco, CA, USA) for 3 min. Mounted sections were subjected to microscopic (EVOS FL Auto, 2.3.3.42, Life technology, Calsbad, CA, USA) observation. Measurement of the thickness of the epidermis was carried out with ImageJ (version 1.53e, Java 1.8.0, National Institutes of Health, Bethesda, MD, USA, https://imagej.nih.gov/ij/download.html, accessed on 24 July 2022).

### 2.5. Immunohistochemical Staining

Paraformaldehyde fixed, paraffin-embedded sections (4 µm in thickness) were baked at 60 °C for 2 h, and a series of deparaffined and hydration processes were conducted. Antigen was retrieved with Tris-ethylenediaminetetraacetic acid (Tris-EDTA) buffer, pH 9.0, at 100 °C for 5 min. The incubation of the anti-Ki67 primary antibody (cat# PRM325AA, 1:5 dilution, BIOCARE MEDICAL, Pacheco, CA, USA) was conducted at 4 °C overnight. Immunohistochemical staining was carried out according to the manufacturer’s instructions (cat# PK-6101 and cat# SK-4100, VECTOR LABORATORY, Newark, CA, USA). All images were captured with a PCOpanda digital camera (sCMOS technology, Fuzhou, China), and a Nikon microscope (Eclipse E800, Melville, NY, USA). QuPath software (version 0.2.3, University of Edinburgh, Edinburgh, UK, https://qupath.github.io/ accessed on 24 July 2022) [32] was used to count Ki67^+^ basal keratinocytes in an unbiased fashion.

### 2.6. Cell Culture and Reagents

Human normal keratinocyte cell line HaCaT (American Type Culture Collection, ATCC, Manassas, VA, USA) and human squamous-cell carcinoma cell line A431 (kindly gifted from Dr. Shitao Li at Tulane University) were maintained in high glucose Dulbecco’s Modified Eagle Medium (DMEM) (cat# 25-500, Genesee Scientific, San Diego, CA, USA) supplied with 10% fetal bovine serum (FBS, cat# 30-2022, ATCC, Manassas, VA, USA). An incubator was set up to maintain a humidified condition at 37 °C with 5% carbon dioxide.

### 2.7. Colony Formation Assay

A total of 5 × 10^5^ HaCaT cells or 3 × 10^5^ A431 cells were seeded into a 6-cm dish 24 h before transfection. A total of 50 nM of control siRNA or siIL17RA (cat# sc-40037, Sant Cruz Biotechnology, Dallas, TX, USA) were delivered into the cells with jetPRIME transfection reagent (cat# 101000046, Polyplus, Illkirch, France). A total of 24 h post-transfection, 2000 transfected cells were placed into 6-well plates supplied with 2 mL of DMEM medium (10% FBS). After 24 h, treatment of centrinone with the final concentration of 0, 4, and 8 µM was conducted, and the medium was replaced every 2 days. A total of 14 days post the initiation of the centrinone treatment, cells were washed with pre-chilled phosphate saline buffer, fixed with 4% paraformaldehyde for 30 min at room temperature, and stained with crystal violet (0.1% in 20% methanol) for 30 min at room temperature. Colonies with more than 50 cells were counted.

### 2.8. Western Blot

Cells were lysed by radioimmunoprecipitation assay (RIPA) buffer (50 mM sodium fluoride, 0.5% NP-40, 10 mM sodium phosphate monobasic, 150 mM sodium chloride, 25 mM Tris pH8.0, 2 mM ethylenediaminetetraacetic, and 0.2 mM sodium vanadate) with a fresh supplement of 1× protease inhibitor cocktail (cat# P8849, Millipore Sigma, St. Louis, MO, USA). Around 80 µg of whole cell lysate was loaded into sodium dodecyl sulfate-polyacrylamide gel electrophoresis (SDS-PAGE) and transblotted to the polyvinylidene fluoride membranes (PVDF, cat# 83-646R, Genesee Scientific, San Diego, CA, USA). Incubation of primary antibodies (anti-GAPDH (cat# AB2302, 1:5000 dilution, Millipore Sigma, St. Louis, MO, USA), anti-IL17RA (cat# sc-376374, 1:500 dilution, Santa Cruz Biotechnology, Dallas, TX, USA), anti-PLK4 (cat# 12952-1-AP, 1:1000 dilution, Proteintech, Rosemont, IL, USA)) was conducted at room temperature for 1.5 h. The IRDye^®^ 680RD Goat anti-Mouse IgG Secondary Antibody (Li-COR, Lincoln, NE, U.S.A) was incubated at room temperature for 45 min with a dilution ratio of 1:10,000. Blots were scanned with an Li-COR Odyssey 9120 Digital Imaging system.

### 2.9. Real-Time Quantitative Reverse Transcription-Polymerase Chain Reaction (qRT-PCR)

Mouse skin samples were ground in liquid nitrogen, and a proper amount of Trizol reagent (cat# 15596018, Ambio, Carlsbad, CA, USA) was used to isolate total RNA, according to the manufacturer’s instructions. The quantification of total RNA was carried out with a microplate spectrometer (Synergy H1, BioTeck, Santa Clara, CA, USA). cDNAs were then synthesized using 500 ng of total RNA and reverse transcriptase (cat# RR037A, Takara, San Jose, CA, USA). Before doing real-time qPCR, the synthesized cDNAs were diluted 6-fold with DNase/RNase-free water. To perform real-time qPCR using QuantStudio 3 (The Applied Biosystems, Waltham, MA, USA), 10 µL of SYBR Green Master mix (cat# A25742, Applied Biosystems, Waltham, MA, USA), 2 µL of diluted cDNA templates, 3 µL of DNase/RNase-free water, and 5 µL of 1 µM primer mixture were mixed. The forward and reverse primers of Mus musculus listed in Table 1 were synthesized using Eurofins Scientific. The real-time qPCR conditions were as follows: 95 °C for 5 min, 40 cycles of 95 °C for 15 sec, and 60 °C for 1 min, then maintain the product at 4 °C. The expression levels of each of our target genes collected from 4 biological replicates were analyzed by the comparative ^∆∆^Ct method [33]. In each sample, GAPDH was used as the internal control, and normal control of WT mouse was set up as a calibrator to normalize all the other groups. Fold change relative to the control group was calculated as 2^−∆∆Ct^ [(^∆∆^Ct = sample (Ct_target_ − Ct_GAPDH_) − calibrator (Ct_target_ − Ct_GAPDH_)].

### 2.10. Bioinformatics Analysis

Original data of GSE109248 were analyzed using the QIAGEN OmicSoft platform (Redwood City, CA, USA). Genes of the PLK family and IL17 family were clustered under the default set of parameters. The transcription levels of those genes were plotted as a heatmap categorized under normal skin, cutaneous lupus erythematosus, and psoriasis. The GEO2R tool was applied to differentially determine the expressed genes between psoriasis and normal skin. The significant genes were plotted as volcano plots using GraphPad Prism (version 9.3.1, San Diego, CA, USA). Pre-organized data downloaded from the GDS2200 database were used to create a heatmap using GraphPad Prism (version 9.3.1), where expression patterns of IL17 family members and PLK family members among normal skin, actinic keratosis (AK), and squamous cell carcinoma (SCC) were described. Data from GSE2503 were used to determine significant genes between SCC and normal skin. The significantly expressed genes were plotted into volcano plots using GraphPad Prism (version 9.3.1). The Human Protein Atlas (HPA) database (https://www.proteinatlas.org/, accessed on 12 July 2022) [34,35,36] contains up-to-date clinical data demonstrating human protein expression patterns among normal and various disease conditions. Immunohistochemical staining of human tissues from the HPA database were downloaded to show the expression pattern of IL17RA (cat# MAB1771, R&D system, Minneapolis, MN, USA), IL17RC (cat# AF2269, R&D system, Minneapolis, MN, USA) and PLK4 (cat# HPA035026, Millipore Sigma, St. Louis, MO, USA) between normal skin and SCC.

### 2.11. Statistical Analysis

Colony formation assay was independently repeated 3 times. After counting colonies containing more than 50 cells, quantification was performed using GraphPad Prism (version 9.3.1). A two-tailed Student’s *t* test was used to evaluate significance. To assess the proliferation rate of basal keratinocytes, 5 male mice and 5 female mice of each genotype were randomly chosen to conduct IHC staining of Ki67. The percentage of Ki67^+^ keratinocytes was counted by QuPath software, followed by a two-tailed Student’s *t* test using GraphPad Prism (version 9.3.1). Microsoft365 Excel was utilized to calculate tumor multiplicity, tumor incidence, and tumor volume to describe tumor burden. A statistical analysis of tumor burden was conducted with GraphPad Prism (version 9.3.1), in which a two-tailed Student’s *t* test and a two-way analysis of variance (ANOVA) were performed, respectively. One-way ANOVA with Tukey’s multiple comparison test was applied to determine the statistical significance of mRNA alteration in qRT-PCR analysis.

## 3. Results

### 3.1. Overexpression of IL17 and PLK Family Members in Psoriasis and SCC

The histopathological disorder of skin autoimmune diseases such as psoriasis and cutaneous lupus erythematosus includes the hyperproliferation of keratinocytes, and the same aberration of keratinocytes has been observed in SCC. To determine common alteration of genes among these disease conditions, we searched the GEO database and explored expression patterns of IL17 and PLK family members. Analysis of original data of GSE109248 using OmicSoft platform (QIAGEN, Redwood City, CA, USA) demonstrated that unlike other PLK family members, the transcription level of PLK4 was relatively higher in psoriasis and lupus compared with normal control (Figure 1A). Comparing PLK4 expression levels between psoriasis and normal skin, we found that PLK4 was significantly increased, suggesting that hyperplasia of the epidermis of skin might be linked with PLK4 overexpression (Figure 1C). IL17 signaling is a well-known signaling provoking the proliferation of keratinocytes in lupus and psoriasis. Monoclonal antibodies against IL17 and IL17RA have been approved for the treatment of psoriasis in clinical practice. We found that transcription levels of IL17A, IL17C, and IL17F were increased in psoriasis compared with normal skin, which was in parallel with their receptors IL17RA, IL17RC, IL17RD, and IL17RE. However, expression levels of IL17B, IL17D, and IL17RB showed the opposite trend (Figure 1B). In the analysis of significantly expressed genes between psoriasis and normal skin, IL17RC was considered as the most significantly overexpressed gene among the IL17 family members, followed by IL17RD, IL17RA, IL17F, and IL17RE (Figure 1C). However, when we explored the expression patterns of IL17 and PLK4 family members among normal skin, actinic keratosis (AK), and cutaneous SCC, none of the IL17 family members showed significant alteration at transcription levels, except that PLK1 was statistically increased in SCC compared with normal skin (Figure 1D,E). Even though no pronounced differences of those genes were found at transcription levels, we expanded our search at the protein expression level. The protein expression levels in many normal and abnormal human tissues were accessible from the Human Protein Atlas (HPA) database. We explored the protein expression patterns of IL17RA, IL17RC, and PLK4. IL17RA was not expressed in the epidermis of normal skin or in invasive cancer cells of SCC. Nevertheless, IL17RC and PLK4 were moderately stained in both conditions (Figure 1F). This expression pattern implies that IL17RC and PLK4 may play a role in SCC.

### 3.2. Effect of IL17RA Knockdown and Centrinone on Squamous Skin Cancer Cells

IL17RA is ubiquitously expressed and heterodimerized with IL17RC to function as a binding complex for IL17A and IL17F. Moreover, IL17RA also dimerizes with IL17RD and IL17RE. Therefore, blocking IL17RA is assumed to dominantly suppress IL17 signaling. To examine the combined effect of blocking IL17 signaling and the inhibition of centriole duplication on the proliferation of keratinocytes, endogenous IL17RA was knocked down with small interference RNA (siRNA), and centrinone was applied to normal human keratinocyte HaCaT and human squamous-cell carcinoma cell line A431 (Figure 2A). The knockdown of endogenous IL17RA in HaCaT and A431 inhibited the formation of cell colonies, especially in A431 cells. Statistical analysis demonstrated that the cancer cell line was more susceptible to decreased expression of IL17RA. However, 4 µM and 8 µM of centrinone completely abolished the cell proliferation of both HaCaT and A431 cell lines (Figure 2B,C).

### 3.3. Knockout of Il17rc Alleviates Skin Lesion Induced by Imiquimod

An imiquimod-induced psoriasis model was established in WT, *Il17ra*(T779A)-KI, and *Il17rc*-KO mouse strains. A total of 25 µg of imiquimod cream was smeared on the shaved back skin of the mice constitutively for 5 days, and after a 2-day interval, mice were euthanized, and the skin samples were collected. The topical treatment 4 µM of centrinone was conducted to determine its inhibition effect on the epidermis. In the duration of treatment, skin lesions were monitored every day (Figure 3A). In WT mice, characteristic lesions, such as pink to salmon-colored plaques, silver-white scales, and bleeding points, were observed on the back skins of both the left and right sides. However, mutation of Threonine 779 of *Il17ra* into Alanine (T779A) mitigated the severity of the skin lesions, and the knockout of *Il17rc* significantly alleviated the skin lesions (Figure 3A). Histopathological examination found that the thickness of the epidermis, including the stratum granulosum, stratum spinosum, and stratum basalis, was dramatically decreased in the *Il17rc*-KO mice (Figure 3B,C). However, the knockin of T779A of *Il17ra* did not affect the hyperplasia of the epidermis in imiquimod-induced psoriasis model, and these changes were not affected by centrinone treatment (Figure 3B,C).

### 3.4. Knockout of Il17rc Inhibits Keratinocyte Proliferation Induced by Imiquimod

To further evaluate the effect of IL17 signaling on keratinocyte proliferation, 5 samples of male mice and 5 samples of female mice of each subgroup were randomly selected from 69 mice. Ki67 is an established prognostic and predictive indicator of cellular proliferation, so immunohistochemical staining of Ki-67 was carried out to assess the proliferation of keratinocytes in this psoriasis model. Compared with WT and *Il17ra*(T779A)-KI mice, *Il17rc*-KO significantly decreased the percentage of Ki67^+^ basal keratinocytes (Figure 4A,B). Similarly, the mutation of T779A of *Il17ra* significantly reduced the proportion of Ki67^+^ basal keratinocytes compared with WT mice (Figure 4A,B). To assess if there is any gender difference regarding the incidence and severity of psoriasis, we further analyzed the differences between Ki67^+^ basal keratinocytes in male and female mice separately. In female mice, *Il17rc*-KO significantly decreased imiquimod-induced Ki67^+^ basal keratinocytes compared with WT and *Il17ra*(T779A)-KI mice (Figure 4C). However, in male mice, we only observed reduction of Ki67^+^ basal keratinocytes in *Il17rc*-KO mice compared with WT mice (Figure 4C).

### 3.5. CEBPβ -Mediated Downstream Signaling Is Inhibited in Psoriasis Model

To investigate the molecular mechanisms under the phenotypes in our psoriasis model, we applied a qRT-PCR assay to evaluate expression levels of C-X-C motif chemokine ligands (CXCL), C-C motif chemokine ligands (CCL), interleukin 6 (IL6), tumor necrosis factor α (TNFα), and CCAAT enhancer binding protein beta (CEBPβ). Although it is not consistent with phenotypic change, CXCL5 was the only CXCL family member that was significantly induced by imiquimod in the *Il17ra*(T779A)-KI mice, and centrinone antagonized this effect. Compared with WT and *Il17rc*-KO mouse strains, the levels of CXCL5 in *Il17*ra(T779A)-KI were significantly higher, while CXCL1, CXCL2, and CXCL9 did not show any significant differences (Figure 5A–D). For the CCL family members, CCL5 was observed to be elevated in *Il17rc*-KO mouse strain with imiquimod treatment, while CCL2, CCL7, and CCL20 did not show any differences (Figure 5E–H). In the WT mouse strain, imiquimod did not induce expression of IL6, but when combined with centrinone, IL6 was significantly increased. However, neither imiquimod nor a combination of imiquimod and centrinone increased the levels of IL6 in *Il17ra*(T779A)-KI and *Il17rc*-KO mouse strains (Figure 5I). The expression of TNFα was only observed to be decreased in the *Il17rc*-KO mouse strain compared with the *Il17ra*(T779A)-KI mouse strain (Figure 5J). In the WT mice, the expression of CEBPβ was dramatically increased by imiquimod, yet it was significantly decreased in the *Il17ra*(T779A)-KI and *Il17rc*-KO mouse strains (Figure 5K).

### 3.6. Mutation of T779A of Il17ra and Knockout of Il17rc Prevent the Development of Skin Papilloma

Following a standard protocol, skin papilloma was initiated by one dose of the carcinogen DMBA and constitutively promoted through topical treatment of TPA twice a week for 13 weeks. Centrinone was topically applied to the right back skin, with solvent control on the left back skin (Figure 6A). Papilloma was estimated to occur after about 6 to 12 weeks. Since our study was to determine the effects of IL17 signaling and centrinone on the initiation and early progression of skin papilloma, this experiment was terminated after the 13th week. During the 13-week monitoring of the mice for skin tumorigenesis, the formation of skin papilloma in WT mice occurred 1 week earlier than in the *Il17ra*(T779A)-KI mice. Tumor incidence in WT mice was also significantly higher than in the *Il17ra*(T779A)-KI mice. The knockout of *Il17rc* completely abolished the formation of skin papilloma. However, topical administration of centrinone made no difference in the development of skin papilloma (Figure 6B,C). Furthermore, tumor multiplicity, which refers to the average number of papillomata per mouse, of each sub-group was monitored until the maximum number of papillomata was consistently achieved. With respect to male mice, without topical treatment of centrinone, the knockout of *Il17rc* significantly inhibited tumor development compared with WT, but *Il17ra*(T779A)-KI did not show the same effect. However, topical administration of centrinone enhanced the reduction in tumor multiplicity in *Il17ra*(T779A)-KI mice compared with WT. The same effect was also observed in *Il17rc*-KO mice (Figure 6D). However, in female mice, the overall tumor multiplicity was less than in male mice. *Il17ra*(T779A)-KI did not alter tumor multiplicity, but *Il17rc*-KO significantly alleviated the growth of tumors (Figure 6D). Tumor volume was another indicator used to describe tumor burden in this study. Two formulas were used to calculate tumor volume based on whether the depth of tumor was measurable (≥1 mm) or not. Because of gender difference of tumor burden, statistical analysis of tumor burden was performed within each gender group. In male mice, tumor volume was not significantly reduced in *Il17ra*(T779A)-KI mice compared with WT mice, but centrinone treatment enhanced this reduction. However, such an effect was not observed in female mice. Consistent with tumor multiplicity, *Il17rc*-KO completely stalled tumor growth without any obvious gender difference (Figure 6E).

### 3.7. Knockout of Il17rc Inhibits Hyperplasia of Keratinocytes in Skin Papilloma Model

The thickness sum of stratum granulosum, stratum spinosum, and stratum basalis was quantified to represent hyperplasia of keratinocytes. In male mice, the proliferation of keratinocytes was hindered by *Il17rc*-KO, but not *Il17ra*(T779A)-KI. Topical administration of centrinone could not inhibit the proliferation of keratinocytes induced by DMBA/TPA (Figure 7A,B). However, an opposite phenotype alteration was observed in female mice, which showed that centrinone treatment favored the reduction of epidermis thickening of *Il17rc*-KO mice compared with *Il17ra*(T779A)-KI mice. However, the difference of keratinocyte proliferation between WT and *Il17rc*-KO did not show any statistical significance (Figure 7A,C). Pronounced tumor formation was observed in WT and *Il17ra*(T779A)-KI mice, but no tumors developed in the *Il17rc*-KO mice (Figure 7D).

### 3.8. CXCL1-Mediated Inflammation Response Is Inhibited in Papilloma Model

In the papilloma model, we evaluated mRNA levels of the same genes examined in the psoriasis model. Except for CXCL1, CCL20, and TNFα, none of the other genes examined showed any differences among WT, *Il17ra*(T779A)-KI, and *Il17rc*-KO mouse strains (Figure 8A–K). In WT mice, CXCL1 was significantly increased by DMBA/TPA treatment. Compared with WT mice, *Il17ra*(T779A)-KI dramatically inhibited the expression of CXCL1. Knockout of *Il17rc* also influenced the expression of CXCL1 in the same trend. However, there was no statistical difference of CXCL1 expression between *Il17ra*(T779A)-KI and *Il17rc*-KO mouse strains. Overall, the treatment of centrinone did not have any effect on the CXCL family members (Figure 8A–D). In addition, we observed that centrinone treatment significantly decreased the expression of CCL20 in WT mice (Figure 8H), and *Il17ra*(T779A)-KI inhibited TNFα expression compared with WT mice when treated with centrinone (Figure 8J).

## 4. Discussion

Our analyses of the GEO and HPA databases showed a relatively high expression of IL17 and PLK family members at the transcription and protein levels in psoriasis and SCC. We found that the inhibition of IL17RA and PLK4 significantly blocked the proliferation of normal keratinocytes and SCC cancer cells in in vitro studies. Our animal models showed that hyperplasia of the epidermis and the proliferation of keratinocytes were decreased by both the T779A mutation of *Il17ra* and the knockout of *Il17rc*. However, topical administration of centrinone to both animal models did not show any alleviation of the disease. Using the imiquimod-induced psoriasis model and the DMBA/TPA-induced skin tumorigenesis model, the present study found that the single knockin mutation of T779A of mouse IL17RA and knockout of mouse IL17RC prevented the development of psoriasis and tumorigenesis in the epidermis. Our findings confirmed the pivotal role of IL17RC in psoriasis [28] and skin tumorigenesis. To our surprise, the T779A mutation of mouse IL17RA did not have the anticipated effects of enhancing the IL17 responses in the skin. Our previous study predicted that the mutation of this IL17RA phosphorylation site might stabilize IL17RA, thus enhancing IL17 inflammatory responses [24]. We speculate that this discrepancy might be caused by the fact that IL17RC plays a more important role in the skin pathogenesis. Another unexpected finding was that centrinone did not have any obvious effects on the skin lesions in in vivo studies, although centrinone abolished cell growth in both normal keratinocytes and the skin cancer cell line in the in vitro studies. A possible reason might be that centrinone did not permeate through the skin to act on the keratinocytes. Further research shall be performed to clarify this.

IL17 signaling plays a pivotal role in many autoimmune diseases (e.g., psoriasis, inflammatory bowel diseases, and spondyloarthropathies) as defense against pathogen insults (e.g., Candida, Cryptococcus, Klebsiella, and Staphylococcus) and human malignancies (e.g., prostate cancer, cervical cancer, esophageal cancer, gastric cancer, hepatocellular carcinoma, lung cancer, non-melanoma skin cancer, and colorectal cancer) [1,21,24,37,38,39,40]. Due to its complex form of receptor dimerization, the function of IL17 signaling is context-dependent, particularly in tumorigenesis [40]. The proliferative signaling of keratinocytes is initiated by IL17 through recruitment of IL-17RA/IL-17RC and then Act1, which is assumed to explain the link between psoriasis and skin tumorigenesis [10,40,41]. Besides blocking IL17 secretion from Th17 cells, an alternative strategy to treat psoriasis and SCC is the targeting of keratinocyte proliferation. Combined pharmacotherapy has been proved to be a good strategy to decrease the possibility of drug resistance [42,43]. Our in vitro colony formation assay supported our hypothesis that the proliferation of the A431 cell line was IL17 signal-dependent, and the skin cancer cells were more sensitive to deprivation of the IL17 signal than normal keratinocytes HaCaT.

In our psoriasis model, skin lesions were induced by the topical administration of imiquimod on the back skin, where hyperproliferation of epidermal keratinocytes in WT mice was persistently induced. In contrast, deficiency of *Il17rc* completely abolished the thickening of the epidermis and the proliferation of keratinocytes, which verified the key role of IL17RA/IL17RC complex in the development of psoriasis. Threonine 779 of mouse IL17RA was reported to be a phosphorylation site of glycogen synthase kinase 3 beta (GSK3β) [24]. We previously predicted that the phosphorylation of IL17RA leads to ubiquitin-proteasome-mediated degradation; thus, the mutation of T779 may elevate the inflammatory response involving IL17 signaling [24]. To test this prediction, we generated the *Il17ra*(T779A)-KI mouse strain and used imiquimod to induce psoriasis. To our surprise, our data did not support our prediction that the T779A mutation alleviated skin lesions. However, we observed that the T779A mutation did increase Ki67^+^ keratinocytes in untreated normal mice. This discrepancy is difficult to explain with the current data and shall be investigated in future studies. Regarding keratinocytes, several studies have implied that chemokines, such as CXCLs, CCLs, IL6, and TNFα, participate in IL17-driven inflammatory responses [1,40,44,45,46]. Our data showed that CXCL5 was the only downstream chemokine that was significantly induced by imiquimod in *Il17ra*(T779A)-KI mice, and treatment with centrinone antagonized this effect. In the obese condition, CXCL5 synergized endoplasmic reticulum stress to exacerbate psoriasis symptoms [47]. Increased level of insulin in obesity stabilized human IL17RA via decreasing the phosphorylation at T780 [24]. In our *Il17ra*(T779A)-KI mouse strain, even though CXCL5 was elevated by imiquimod, skin lesions were similar to those in WT mice. Therefore, CXCL5 might not play a key role in this model. IL6 and TNFα are common IL17-downstream cytokines in autoimmune environments [45]. With stimulation by imiquimod, TNFα was significantly higher in *Il17ra*(T779A)-KI mice than in *Il17rc*-KO mice, which supports TNFα as an IL17-downstream cytokine. The combination of imiquimod and centrinone significantly increased IL6 expression in WT mice, but not in *Il17ra*(T779A)-KI and *Il17rc*-KO mice. It is plausible that an inflammatory response involving IL6 is suppressed in *Il17ra*(T779A)-KI and *Il17rc*-KO mice. The mechanisms of how centrinone induced IL6 expression are unknown. CEBPβ expression was decreased in *Il17ra*(T779A)-KI and *Il17rc*-KO mouse strains, confirming CEBPβ as a key transcription factor downstream of IL17 signaling [45].

Previous reports have demonstrated that IL17 signaling was ubiquitously functional in a wide range of tissues and pathological conditions [40,45]. By collaboration with other inflammatory molecules and transcription factors, IL17 signaling has context-specific contributions to tumorigenesis, which have been poorly appreciated [40]. The mutation of p53 that increases RAS signaling and decreases Notch signaling is a common contributor to SCC transformation [18]. DMBA predominantly targets basal keratinocytes to induce a transversion of adenine 182 into thymine, and nucleated keratinocytes are prompted to expand colonies, resulting in the thickening of the epidermis [30]. Tumor development in the DMBA/TPA-induced skin cancer was enhanced by IL17, and high levels of IL17 in the microenvironment promoted SCC formation [48]. Here, we utilized WT, *Il17ra*(T779A)-KI, and *Il17rc*-KO mouse strains to establish a skin carcinogenesis model to determine the effect of combining the destruction of IL17RA/IL17RC dimer and the topical administration of centrinone on skin tumorigenesis. *Il17ra*(T779A)-KI significantly reduced tumor incidence, and *Il17rc*-KO completely prevented the development of skin tumors, which supports the view that IL17 signaling promotes tumorigenesis [40]. In addition, we also noted obvious gender differences in tumor multiplicity and tumor volume. In the WT mice, although statistical significance was not reached, the tumor burden in male mice was higher than female mice. The treatment with centrinone slightly exacerbated tumor burden in WT male mice, but the condition was improved by *Il17ra*(T779A)-KI and *Il17rc*-KO. This gender difference is in line with the clinical fact that women usually have less severe psoriasis [49], indicating an underlying shared mechanism regulating the proliferation of keratinocytes in SCC and in the psoriasis condition. Even though the proliferation of the epidermis in both male and female mice was significantly induced by DMBA/TPA among WT, *Il17ra*(T779A)-KI, and *Il17rc*-KO mouse strains, *Il17ra*(T779A)-KI alleviated tumor formation and *Il17rc*-KO totally abolished it. This was similar to what we observed in the psoriasis model, and we do not know why *Il17ra*(T779A)-KI alleviates instead of enhancing skin tumors, as predicted by our previous study [24]. The induction of CXCL1 was one of the hallmarks of IL17 [40,45], and the change of CXCL1 expression was in line with the alteration of the phenotype in our DMBA/TPA-induced model, supporting the critical role of CXCL1 in the IL17-driven skin carcinogenesis. CCL20 was reported to promote cancer by directly enhancing cancer cell proliferation or indirectly remodeling the tumor microenvironment [50]. Although CCL20 was inhibited by centrinone, hyperplasia of the epidermis in WT mice was not decreased, indicating that CCL20 may not be a critical player in skin tumorigenesis. Due to huge individual variations among the samples, we were unable to find significant changes among other genes examined. However, data collected from this DMBA/TPA model were consistent with a previous report that *Il17^-/-^* reduced skin tumorigenesis [48].

With respect to limitations of our study, they mainly occur in three aspects: (1) The topical administration of centrinone did not show any inhibition of basal keratinocyte proliferation in in vivo studies, which was contradictory to our colony formation experiments. However, we did not perform any studies to reveal the possible mechanisms; for example, whether centrinone was permeated through the skin. Future experiments should be performed to study the delivery of this drug. (2) The RNA-sequencing approach was not used to determine transcriptomic profiling, and instead, only a few known IL17-downstream genes were examined, and limited significant players were identified. Therefore, the dominant participant(s) in the formation of skin lesions in our model remain obscure. (3) The susceptibility of mouse strains to TPA is in the decreasing order of SENCAR > DBA/2 > CD-1 > C3H/He >> C57BL/6J [30]. We used mice with a C57BL/6J background, which are least sensitive to TPA, because our mutant mice are of C57BL/6J background. This could be one of the reasons that we did not observe development of invasive skin cancer. The failure to develop invasive skin cancer might also be caused by the short 14 weeks of follow-up. Future studies should consider a more sensitive genetic background and a longer follow-up time.

## 5. Conclusions

In summary, we found that the proliferation of keratinocytes was not stalled by the PLK4 inhibitor centrinone, but was inhibited by *Il17rc*-KO. *Il17ra*(T779A)-KI significantly inhibited skin papilloma formation, but only slightly decreased epidermal thickening in the psoriasis model. Our study has confirmed the important role of the IL17RA/IL17RC heterodimer in promoting keratinocyte proliferation in psoriasis and skin papilloma models. Therapies that target IL17RC may be developed as potential therapeutic approaches to inhibit psoriasis and skin carcinogenesis.

## Figures and Tables

**Figure 1 biomedicines-10-01976-f001:**
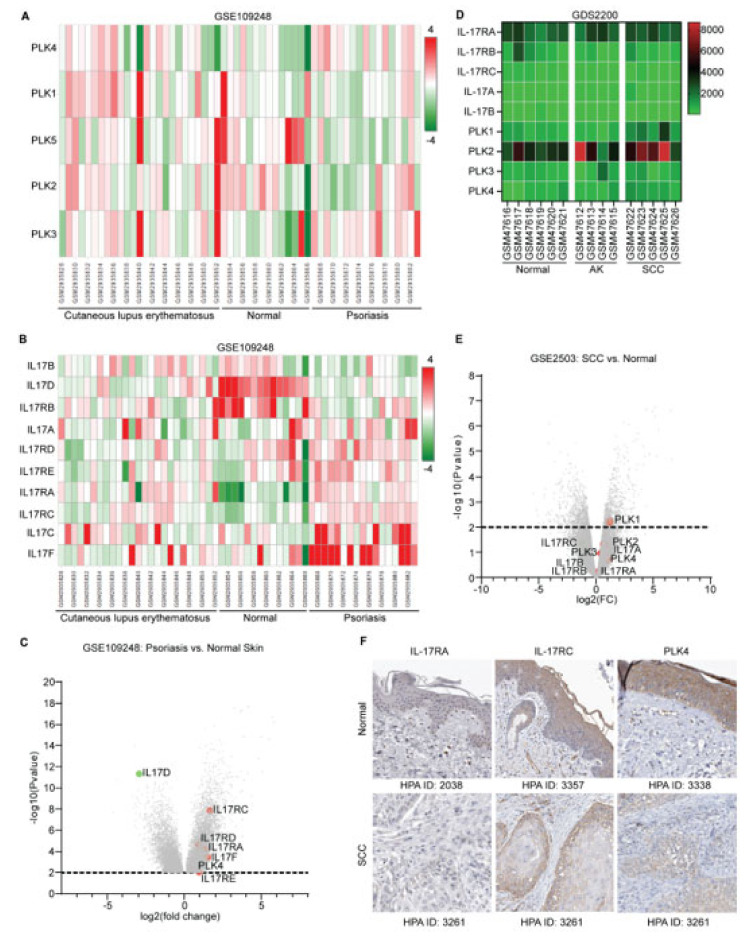
Expression pattern of IL17 and PLK family members in skin cancer and autoimmune diseases. (**A**,**B**) Original data of GSE109248 were analyzed using the OmicSoft platform. The heatmap shows the expression patterns of PLK family members among normal skin, cutaneous lupus erythematosus, and psoriasis. (**C**) The GEO2R tool was applied to differentially determine the expressed genes (*p* value ≤ 0.01) between psoriasis and normal skin. (**D**) The heatmap shows expression patterns of IL17 family members and PLK family members among normal skin, actinic keratosis (AK), and squamous cell carcinoma (SCC) of skin. Data were downloaded from GDS2200 database. (**E**) The volcano plot demonstrates significant genes between SCC and normal skin. Data were downloaded from GSE2503. (**F**) Immunohistochemical staining of human tissues from The Human Protein Atlas (HPA, https://www.proteinatlas.org/, accessed on 12 July 2022) shows expression patterns of IL17RA, IL17RC, and PLK4 between normal skin and SCC.

**Figure 2 biomedicines-10-01976-f002:**
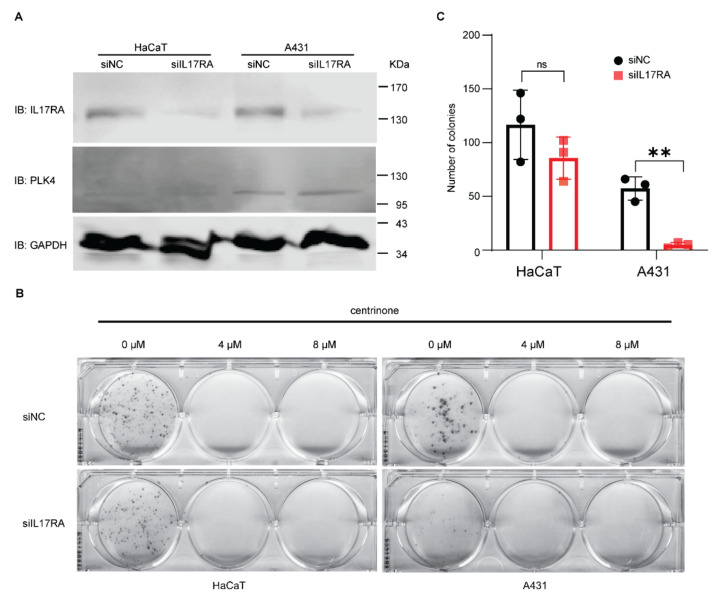
Effect of IL17RA knockdown and centrinone on squamous skin cancer cells. (**A**) Endogenous IL17RA of HaCaT and A431 was knocked down using small interference RNA of IL17RA. (**B**,**C**) Colony formation assay was conducted to determine the effect of IL17RA knockdown and inhibition of PLK4 on the proliferation of normal skin cell line HaCaT and squamous skin cancer cell line A431. A two-tailed Student’s *t* test was applied to analyze statistical significance. Error bar represents mean ± standard deviation (SD); ns—no significance; ** —*p* < 0.01.

**Figure 3 biomedicines-10-01976-f003:**
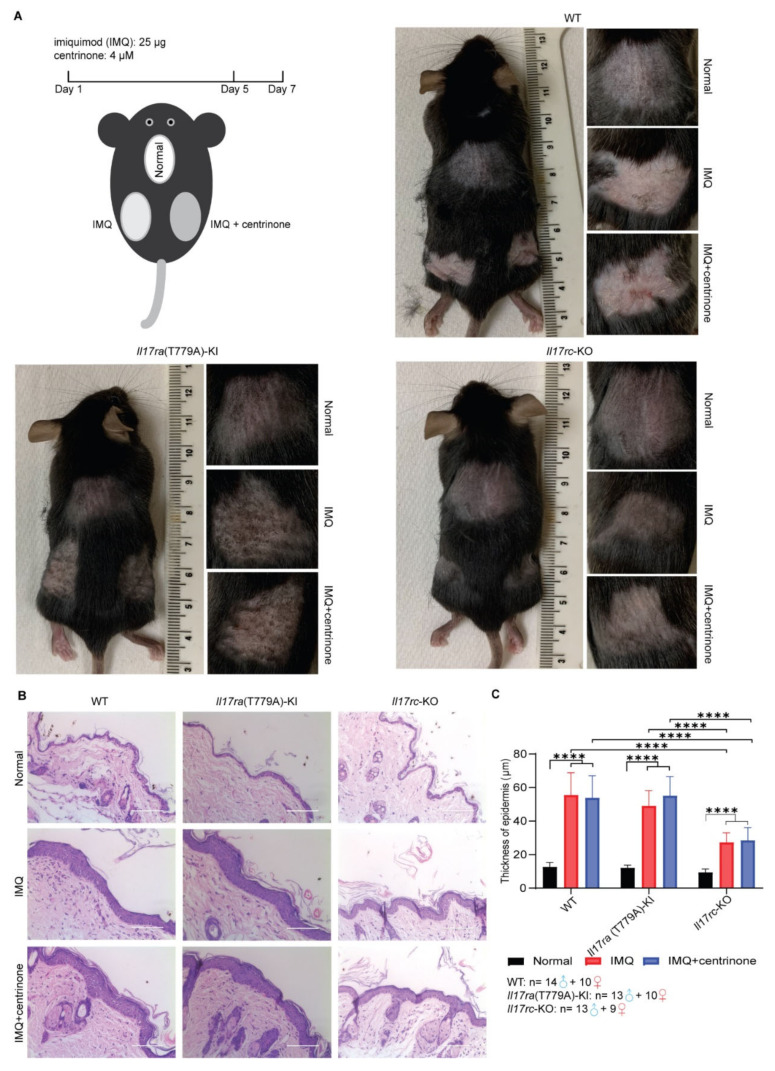
Knockout of *Il17rc* alleviates skin lesions induced by imiquimod. (**A**) Diagram of establishing the imiquimod-induced psoriasis mouse model and skin lesions of WT C57BL/6J, *Il17ra*(T779A)-knockin, and *Il17rc*-knockout strains. (**B**) Representatives of hematoxylin and eosin stain of skin lesions. Scale bar represents 100 µm. (**C**) To analyze the statistical significance of the thickness of epidermis, a two-tailed Student’s *t* test was applied. Error bar represents mean ± SD. **** indicates *p* < 0.0001. The numbers of male and female mice in each group are indicated in the panel.

**Figure 4 biomedicines-10-01976-f004:**
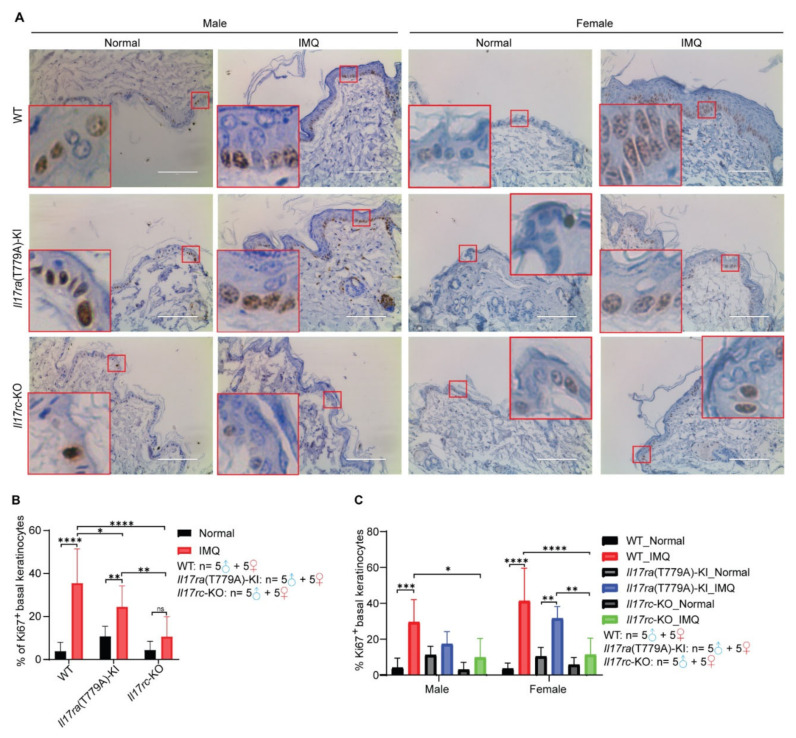
Knockout of *Il17rc* inhibits keratinocyte proliferation induced by imiquimod. (**A**) Representative of immunohistochemical staining of Ki67 in the epidermis of skin lesions. Skin samples of 5 male mice and 5 female mice of each subgroup were used for immunohistochemical staining. Representative areas showing basal keratinocytes were zoomed in. Scale bar represents 100 µm. (**B**,**C**) Percentage of Ki67^+^ basal keratinocytes were counted, and statistical significance was assessed with a two-tailed Student’s *t* test. Scale bar represents 100 µm. Error bar represents mean ± SD; ns—no significance; *— *p* < 0.05; **— *p* < 0.01; ***— *p* < 0.001; ****— *p* < 0.0001. The numbers of male and female mice in each group are indicated in the panel.

**Figure 5 biomedicines-10-01976-f005:**
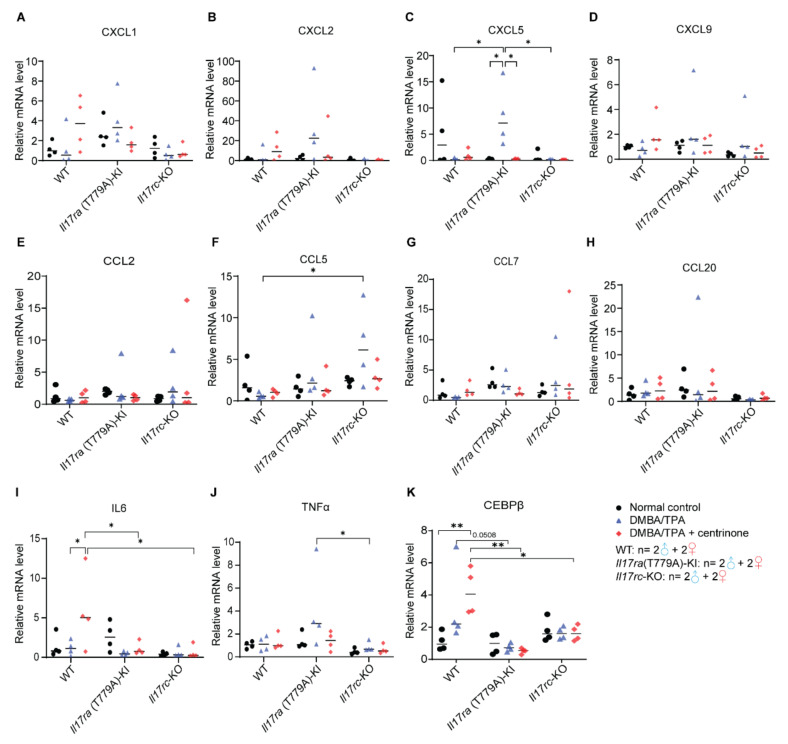
CEBPβ-mediated downstream signaling is inhibited in the psoriasis model. Mouse skin was collected from 4 individuals of each genotype. Total RNA was isolated, and the expression levels of CXCL1 (**A**), CXCL2 (**B**), CXCL5 (**C**), CXCL9 (**D**), CCL2 (**E**), CCL5 (**F**), CCL7 (**G**), CCL20 (**H**), IL6 (**I**), TNFα (**J**), and CEBPβ (**K**) were evaluated using real-time PCR. Normalized value was presented as mean ± SD. Statistical significance was assessed by one-way ANOVA with Tukey’s multiple comparison test. *—*p* < 0.05; **—*p* < 0.01. The numbers of male and female mice in each group are indicated in the panel.

**Figure 6 biomedicines-10-01976-f006:**
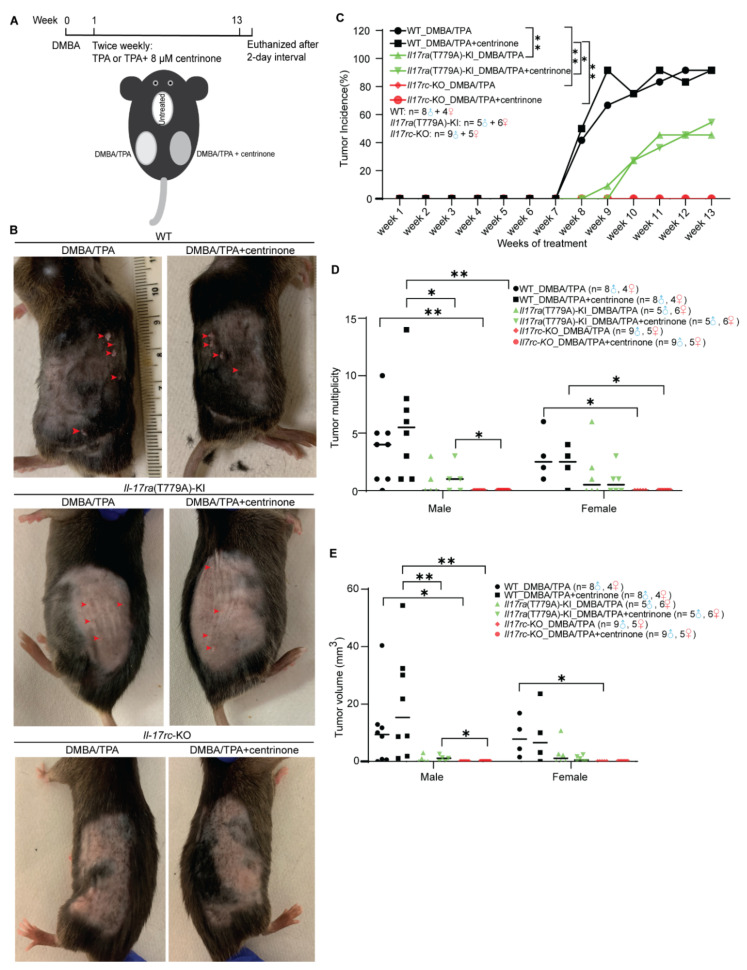
Mutation of T779A of *Il17ra* and knockout of *Il17rc* prevent the development of skin papilloma. (**A**) Diagram of establishing DMBA/TPA-induced two-stage skin papilloma mouse model. (**B**) Representative of tumors or nodules of WT, *Il17ra*(T779A)-knockin, and *Il17rc*-knockout C57BL/6J mouse strains. Arrowheads indicate papillomata (≥1mm) and nodules (<1 mm). (**C**–**E**) Plots demonstrate tumor incidence, tumor multiplicity, and tumor volume. Statistical significance was assessed with a two-tailed Student’s *t* test and two-way ANOVA. Error bar represents mean ± SD. *— *p* < 0.05; **— *p* < 0.01. The number of male and female mice in each subgroup are indicated in the panel.

**Figure 7 biomedicines-10-01976-f007:**
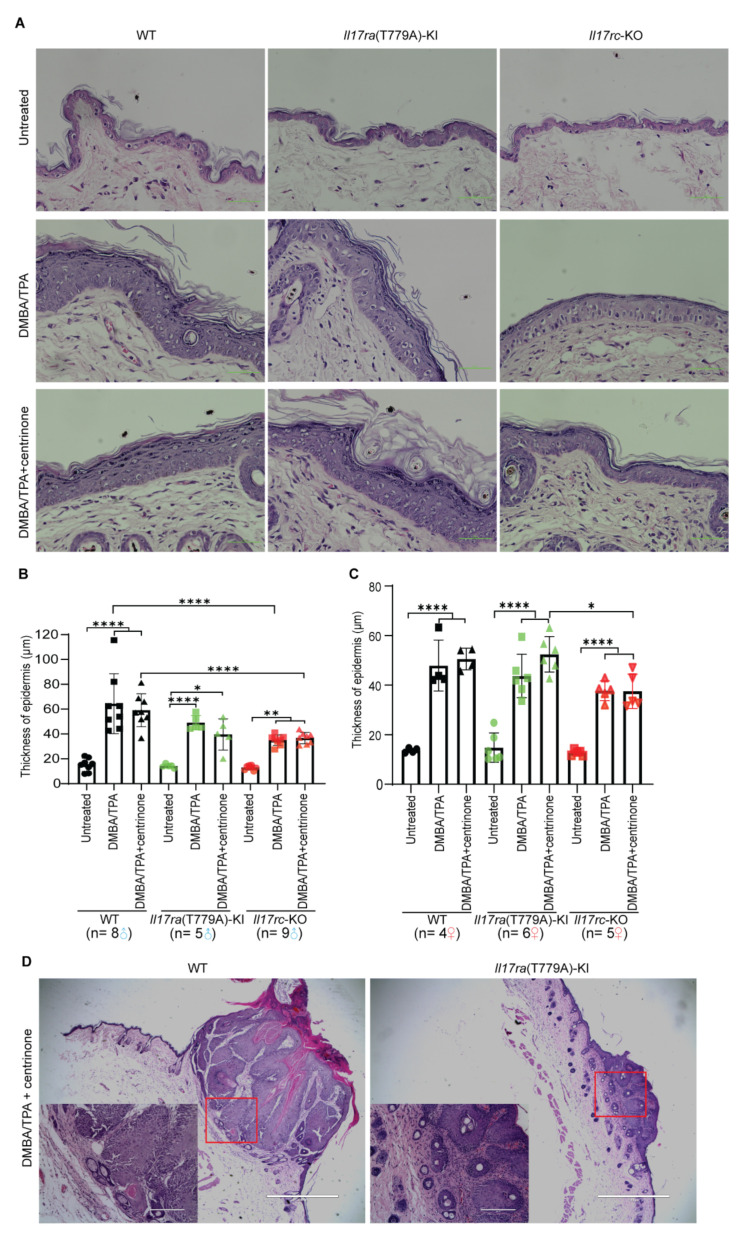
Knockout of *Il17rc* stalls proliferation of keratinocytes in the skin papilloma model. (**A**) Representatives of hematoxylin and eosin staining of skin epidermis. Scale bar represents 50 µm. (**B**,**C**) Statistical analysis of the thickness of the epidermis was carried out by a two-tailed Student’s *t* test. (**D**) Representative of hematoxylin and eosin stain of skin tumor. Scale bar represents 1000 µm (low magnification) and 200 µm (high magnification). Error bar represents mean ± SD. *—*p* < 0.05; **—*p* < 0.01; ****—*p* < 0.0001. The numbers of male and female mice in each group are indicated in the panel.

**Figure 8 biomedicines-10-01976-f008:**
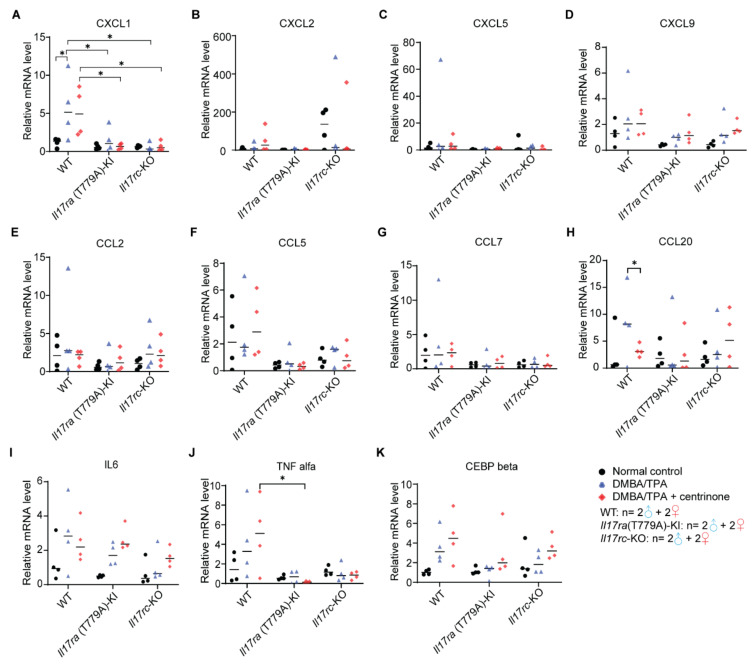
CXCL-mediated inflammatory response is inhibited in the papilloma model. Mouse skin was collected from 4 individuals of each genotype. Total RNA was isolated and the expression levels of CXCL1 (**A**), CXCL2 (**B**), CXCL5 (**C**), CXCL9 (**D**), CCL2 (**E**), CCL5 (**F**), CCL7 (**G**), CCL20 (**H**), IL6 (**I**), TNFα (**J**), and CEBPβ (**K**) were evaluated by real-time PCR. Normalized value was presented as mean ± SD. Statistical significance was assessed by one-way ANOVA with Tukey’s multiple comparison test. *—*p* < 0.05. The number of male and female mice in each group are indicated in the panel.

**Table 1 biomedicines-10-01976-t001:** Primers used in real-time qPCR.

Gene Name	Forward Primer	Reverse Primer
mCXCL1	5′-CACCCAAACCGAAGTCATAG-3′	5′-AAGCCAGCGTTCACCAGA-3′
mCXCL2	5′-CGCCCAGACAGAAGTCATAG-3′	5′-TCCTCCTTTCCAGGTCAGT TA-3′
mCXCL5	5′-GGTCCACAGTGCCCTACG-3′	5′-GCGAGTGCATTCCGCTTA-3′
mCXCL9	5′-AATGCACGATGCTCCTGCA-3′	5′-AGGTCTTTGAGGGATTTGTAGTGG-3′
mCCL2	5′-TGTATGTCTGGACCCATTCCT-3′	5′-GCCTGCTGTTCACAGTTGC-3′
mCCL5	5′-CACCACTCCCTGCTGCTT-3′	5′-ACACTTGGCGGTTCCTTC-3′
mCCL7	5′-TTGACATAGCAGCATGTGGAT-3′	5′-TTCTGTGCCTGCTGCTCATA-3′
mCCL20	5′-AACTGGGTGAAAAGGGCTGT-3′	5′-GTCCAATTCCAT CCCAAAAA-3′
mIL6	5′-CTACCCCAATTTCCAATGCT-3′	5′-ACCAC AGTGAGGAATGTCCA-3′
mTNFα	5′-GGTCTGGGCCATAGAACTGA-3′	5′-TCTTCTCATTCCTGCTTGTGG-3′
mCEBPβ	5′-CGCACCACGA CTTCCTCT-3′	5′-CGAGGCTCACGTAACCGT-3′
mGAPDH	5′-TGC ACCACCAACTGCTTAC-3′	5′-GGATGCAGGGATGATGTTC-3′

## Data Availability

Not applicable.

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
