# Peer review of "Defect of IL17 Signaling, but Not Centrinone, Inhibits the Development of Psoriasis and Skin Papilloma in Mouse Models"

_biomedicines, 2022, doi:10.3390/biomedicines10081976_

Round 1

Reviewer 1 Report

The article is about e the cooperation between IL17 signaling 26 and centriole duplication in epidermal proliferation in psoriasi and SCC.

ABSTRACT: please clarify the sentence ". Interleukin-17 (IL17) provokes proliferation of 24 epidermis leading to psoriasis and overexpression of Polo-like kinase 4 (PLK4), which controls cen- 25 triole duplication, has been identified in SCC."

Author wrote about SCC and skin papilloma.The introduction was about pasoriasi and SCC. The experiments were on a model of skin papilloma.  Please clarify this point. 

No further revision required.

Author Response

Point 1: ABSTRACT: please clarify the sentence "Interleukin-17 (IL17) provokes proliferation of epidermis leading to psoriasis and overexpression of Polo-like kinase 4 (PLK4), which controls centriole duplication, has been identified in SCC."

Response 1: We agree that the wording in this sentence is ambiguous and we have rewritten it to improve clarity (page 1, lines 24-27 of the revised manucript).     

Point 2: Author wrote about SCC and skin papilloma.The introduction was about pasoriasi and SCC. The experiments were on a model of skin papilloma.  Please clarify this point.

Response 2: We intended to study SCC, but due to the short observation time, the skin lesions did not progress into invasive skin cancer and stayed as skin papilloma. Skin papilloma is considered the pre-invasive stage of SCC. Thus, we consider citations of SCC in the introduction are appropriate because the previous reported findings were based on SCC.

Reviewer 2 Report

1.  "The global annual incidence of psoriasis is estimated to be 4,622,594

worldwide in 2019 while around 7.55 US adults are estimated to be

bearing psoriasis in 2020."

Is the global incidence of psoriasis the newly diagnosed cases in 2019?

Please clarify this.  Add references.

2.  lines 46-50:  Psoriatics have a higher incidence of squamous cell 

carcinoma.  Is thios, at least in part, attributable to the therapy rather than

to the disease.  Please comment.

3. line 310 "exam" --> examine

4. Figure 7A: The scale bars are missing.

5. lines 491 & 496: Delete skin.

6. line 566: "declined" --> reduced

7.  line 73:  "tumor protein"  -->  tumor supressor protein

Author Response

Point 1.  "The global annual incidence of psoriasis is estimated to be 4,622,594 worldwide in 2019 while around 7.55 million US adults are estimated to be bearing psoriasis in 2020."

Is the global incidence of psoriasis the newly diagnosed cases in 2019?

Please clarify this.  Add references.

Response 1: We realized this ambiguous interpretation and citation of the references. We have clarified it in the revised manuscript and add reference (page 1 and page 2, lines 44-47).

Point 2.  lines 46-50:  Psoriatics have a higher incidence of squamous cell carcinoma.  Is this, at least in part, attributable to the therapy rather than to the disease.  Please comment.

Response 2: To figure out the causal relationship between psoriasis and squamous cell carcinoma is challenging and complicated, but there are many risk factors that potentially contribute to the increased cancer incidence, for example, aging, specific lifestyle, treatment, mental disorder, and inflammatory process. Current treatments of psoriasis include topical therapy, phototherapy, and oral or injected medications. Although phototherapy was reported to increase SCC, studies to explore the association between increasingly used biological therapy and increased SCC incidence are in paucity. In order to address the reviewer’s comment, the introduction section of the revised manuscript includes addional information addressing the attributable role of therapy to the increased cancer incidence.

Point 3. line 310 "exam" --> examine

Response 3:  The word “exam” has been replaced with “examine” in the revised manuscript.

Point 4. Figure 7A: The scale bars are missing.

Response 4:  We recognized the missing scale bars and they are substituted with pictures with scale bars in the revised manuscript.

Point 5. lines 491 & 496: Delete skin.

Response 5:  The word “skin” has been deleted in the revised manuscript.

Point 6. line 566: "declined" --> reduced

Response 6:  The word “declined” has been replaced with “reduced” in the revised manuscript.

Point 7.  line 73:  "tumor protein"  -->  tumor supressor protein

Response 7:  The word “tumor protein” has been substituted with “tumor suppressor protein” in the revised manuscript.